# MoEsturizer: Resource-Efficient MoE Upcycling for Small Language Models

## Abstract

Large language models (LLMs) are typically scaled through billions of parameters and trillions of tokens, making progress largely restricted to organizations with substantial resources. Recent work on Mixture-of-Experts (MoE) upcycling shows that dense pretrained models can be transformed into sparse MoE variants, but prior studies have focused on large models and required extensive additional training. In this work, we demonstrate that MoE upcycling is also effective for small language models (sub-billion parameters) using only a few hundred thousand samples of supervised fine-tuning. Remarkably, upcycled models consistently outperform their dense base models and remain competitive with dense counterparts of equivalent total size, despite activating fewer parameters at inference. Our study highlights MoE upcycling as a lightweight and practical scaling strategy, while providing empirical insights into its efficiency and limitations. These results establish MoE upcycling as a reproducible pathway for enhancing small models under realistic resource budgets, broadening access to language model improvement.

## 1 Introduction

Large language models (LLMs) have achieved remarkable progress by scaling to ever-larger parameter counts and training corpora. However, this trajectory relies on billions of parameters and trillions of tokens, keeping such advances accessible only to organizations with substantial computational resources. For smaller research groups and individuals, the question remains: can existing pretrained models be scaled or improved meaningfully under realistic constraints of data and compute?

Recent studies on Mixture-of-Experts (MoE) upcycling suggest that dense pretrained models can be transformed into sparse MoE variants, thereby increasing effective capacity with limited additional training. Yet, these approaches have largely targeted billion-scale or larger models and assumed access to billions of tokens for adaptation, leaving open whether such methods are viable in the small-model regime.

In this work, we revisit MoE upcycling under an extremely resource-constrained setting. Starting from publicly available instruction-tuned small language models (SLMs, sub-billion parameters), we apply MoE transformation followed by fine-tuning on only a few hundred thousand tokens. Despite this modest scale, we find that upcycled models consistently outperform their dense baselines of comparable size, showing that meaningful improvements are achievable without massive retraining.

Our study makes three contributions:

- Scale shift: We show that MoE upcycling remains effective even when scaling down to smaller models and training on only a fraction of the data typically used in prior work.
- Practicality: By leveraging open instruction-tuned checkpoints and lightweight fine-tuning, our approach is feasible on consumer-grade GPUs and reproducible by individuals.
- Empirical insights: We provide systematic analysis of the impact of expert number, depth scaling, and efficiency–performance trade-offs in the small-model regime.

Together, these findings establish that MoE upcycling is a practical and accessible strategy for enhancing smaller models under realistic budgets. By showing that even a few hundred thousand

tokens of additional fine-tuning are sufficient to boost widely available SLMs, we demonstrate a pathway by which individuals can meaningfully build stronger models with limited resources, broadening participation in language model research.

## 2 RELATED WORK

### 2.1 MIXTURE-OF-EXPERTS (MoE) ARCHITECTURES

Mixture-of-Experts (MoE) increases capacity cost-efficiently by adding many feed-forward experts while activating only a small subset per token (Shazeer et al., 2017). In Transformer LMs, large-scale MoE-FFN was first operationalized in GShard (Lepikhin et al., 2020), and subsequently popularized by Switch Transformers, which simplified training and adopted top-1 gating (Fedus et al., 2022). This line catalyzed variants including GLaM (Du et al., 2021), ST-MoE (Zoph et al., 2022), and Expert-Choice routing (Jiang et al., 2024), where experts select tokens under fixed capacity. Recent sparse-MoE LLMs such as GLaM and Mixtral 8×7B (2024) demonstrate low active-parameter counts at inference ($\approx 13B$ active with $\approx 47B$ accessible) while leveraging much larger total capacity. However, these approaches typically require substantial compute for both pretraining and alignment, making them difficult to adopt in smaller-scale research or business settings.

### 2.2 POST-HOC MoE TRANSFORMATION (MoE UPCYCLING)

An alternative line of work has explored transforming pretrained dense models into MoE structures. MoEfication (Zhang et al., 2021) proposed splitting pretrained feed-forward neurons into expert groups with a lightweight router, preserving most of the original knowledge while reducing inference cost. Sparse Upcycling (Komatsuzaki et al., 2023) extended this concept by duplicating entire FFN blocks into experts, then continuing pretraining. While effective, this required substantial additional training tokens (up to 1T), limiting accessibility. More recently, LLaMA-MoE (Zhu et al., 2024), LLaMA-MoE v2 (Qu et al., 2024), Llama 3 Meets MoE (Vavre et al., 2024) applied similar ideas to the LLaMA family, and Camelidae/PESC (Wu et al., 2024a), UpIT (Hui et al., 2024) proposed a parameter-efficient approach by inserting adapter-based experts during instruction tuning, requiring only small-scale additional training. These studies highlight that MoE upcycling can leverage pretrained knowledge to achieve substantial performance gains without full retraining, but most work has focused on medium-to-large LLMs and still assumes non-trivial compute budgets. Other directions have explored cost-efficient upcycling from dense models, such as refactored MoE variants that restructure dense FFNs (Cai et al., 2024; Pei et al., 2025; Gao et al., 2025) or MoE approaches combined with LoRA (Wu et al., 2024b; Li et al., 2024), but these still require cluster-level computational costs. More recently, fine-grained MoE (Yang et al., 2024; 2025; DeepSeek-AI et al., 2024; 2025; He et al., 2025) has emerged as a new line of work, offering strong performance guarantees; however, because it partitions the hidden dimensions of FFNs into smaller expert units, it typically demands extensive pretraining and post-training to be effective.

## 3 METHODOLOGY

### 3.1 OVERVIEW

Our objective is to upscale pretrained small language models (SLMs) into stronger generative models using **minimal additional resources**. To this end, we propose **MoE upcycling**, where the feed-forward networks (FFNs) in a dense model are transformed into sparse Mixture-of-Experts (MoE) modules. Figure 1 illustrates the overall methodology, contrasting the baseline dense model with the MoE-upcycled variant.

### 3.2 BASE MODEL

We start from a publicly available instruction-tuned SLM with fewer than 1B parameters. The base transformer architecture follows the standard decoder-only design, where each block consists of a multi-head self-attention (MHSA) layer and a feed-forward network (FFN/MLP), repeated across

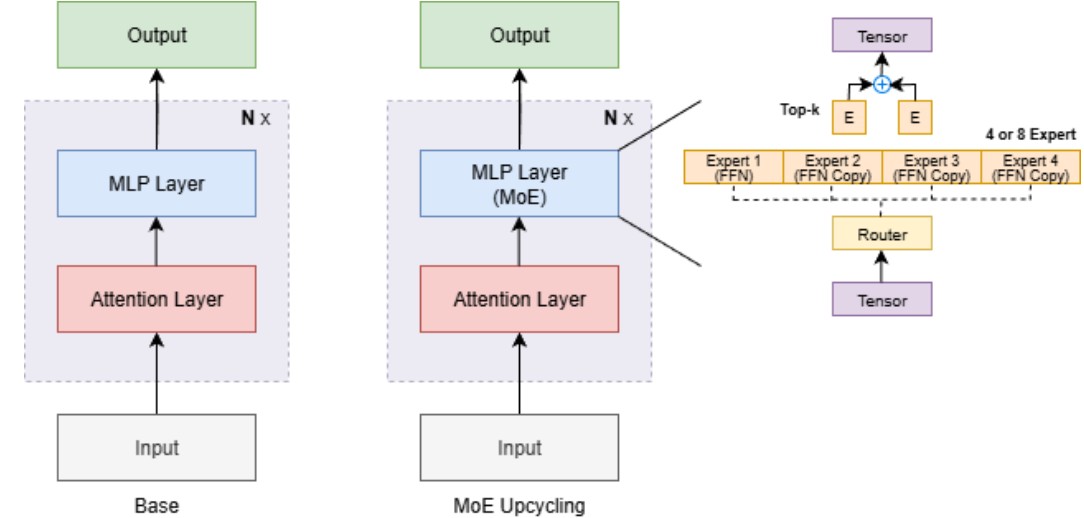

Figure 1: Comparison of three strategies: (left) Base dense model, (middle) MoE upcycling, (right) our proposed hybrid DUS+MoE approach (MoEsturizer).

$N$ layers:

$$h^{(l+1)} = \text{FFN}(\text{MHSA}(h^{(l)})), \tag{1}$$

where $h^{(l)}$ denotes the hidden representation at layer $l$. These pretrained dense models serve as the foundation for our upcycling framework.

### 3.3 MoE Upcycling

The core component of our method is **MoE upcycling**. Each dense FFN block is replaced with a sparse Mixture-of-Experts (MoE) module:

$$y = \sum_{i=1}^{k} G(x)_i \cdot \text{FFN}_i(x), \tag{2}$$

$$G(x) = \text{TopK}(\text{softmax}(Wx)), \tag{3}$$

where $k$ is the number of experts, $\text{FFN}_i$ denotes an expert initialized as a copy of the pretrained FFN, and $G(x)$ is a router function selecting the top-$k$ experts per token. This transformation preserves pretrained knowledge while expanding the model's effective capacity, with only a fraction of experts activated per input.

### 3.4 Lightweight Fine-tuning

To ensure accessibility, we avoid full retraining. Instead, we apply fine-tuning using approximately 150K supervised samples:

- Training can be performed on consumer-grade GPUs.
- Upcycled models consistently outperform their dense base counterparts.
- Despite activating fewer parameters at inference, they remain competitive with dense models of equivalent total size, making them resource-efficient and faster at inference.

This demonstrates that meaningful improvements can be achieved under realistic computational budgets.

**Practical Footprint (Lightweight).** All fine-tuning experiments were conducted on a **single** consumer-grade GPU: **NVIDIA RTX PRO 6000 Max-Q (96 GB VRAM)**. End-to-end wall-clock

time was ~**1.5 hours** for the shortest runs and **under 8 hours** even for the longest setting, making the entire pipeline reproducible within a **single day** on a personal workstation. On the data side, unlike common SFT post-training setups that typically rely on 500k–1M+ samples, we filtered and preprocessed **only ~150k** *publicly available* samples, yet still observed consistent gains. Taken together—*short time, small data, and modest resources*—this evidences the practical *lightweight* nature of METHOD

| Item | Value |
|---|---|
| GPU | NVIDIA RTX PRO 6000 Max-Q (96 GB VRAM) |
| #GPUs | 1 |
| Total SFT samples | ~150k (filtered & preprocessed from public datasets) |
| Wall-clock training time | ~1.5h (shortest) – <8h (longest) |
| Reproducibility | Single-day (within 24h) on a personal workstation |

Table 1: Fine-tuning footprint of METHOD. All results reproduced on a single workstation.

## 4 EXPERIMENTS

### 4.1 TRAINING DATA

**Training corpora and sampling.** We assemble a small, domain-aligned post-training corpus that mirrors our evaluation tasks by combining the following datasets: Tülu 3 SFT mixture *manually filtered by us to English-only and STEM-only*; (Lambert et al., 2024); ARC-Challenge and ARC-Easy `train` (Clark et al., 2018); MMLU `train` (Hendrycks et al., 2021b;a); MMLU-Pro `train` (Zheng, 2024); GSM8K `train` (Cobbe et al., 2021); and HellaSwag `train` (Zellers et al., 2019). For each dataset $D$ with pool size $|D|$, we set a sampling weight $w_D \propto |D|$ and draw i.i.d. examples uniformly within $D$ until reaching a global budget of **150k** examples. This keeps the post-training budget intentionally small while matching evaluation domains.

### 4.2 EVALUATION DATASETS

We evaluate on nine public benchmarks spanning commonsense reasoning, domain knowledge, and STEM problem solving: ARC-Challenge and ARC-Easy (Clark et al., 2018), GPQA-Diamond and GPQA-Main (Rein & et al., 2023), GSM8K (Cobbe et al., 2021), HellaSwag (Zellers et al., 2019), MATH-500 (a subset of the PRM800K math splits; (Lightman et al., 2023)), MMLU (Hendrycks et al., 2021b;a), and MMLU-Pro (Zheng, 2024). Where a benchmark family overlaps with our post-training sources, we use *disjoint* splits (e.g., train vs. test), and we exclude benchmarks that only provide a single split. This suite is widely used, admits standardized automatic scoring, and directly probes whether upscaling small dense LMs to MoE improves their preexisting abilities in knowledge-intensive and STEM-heavy settings.

### 4.3 MAIN RESULTS

**Terminology.** We use the following controlled terms throughout: **Base (IT)** — the publicly available *instruction-tuned* dense model that serves as the starting point for MoE conversion (not a purely pre-trained LM); **Base+SFT (control)** — Base (IT) further trained on our 150k post-training corpus; **MoE+SFT (ours)** — the same Base (IT) converted to MoE and trained on the identical corpus; **Larger Dense (headroom)** — the next larger dense model whose *total* parameters are comparable to the *active* parameters of our MoE; **Larger Dense+SFT (control-L)** — Larger Dense further trained on the same corpus.

**Evaluation protocol.** To ensure fair comparison, all systems are scored *zero-shot, pass@1* with same configurations of generations; no chain-of-thought, self-consistency, or tool use is allowed. To isolate structural gains from data effects, we report Base+SFT (control) alongside MoE+SFT (ours). Training is limited to **one epoch** for both control and MoE, with a **fixed step budget of 1056 updates** (derived from a 150k example budget with **10% held out for validation**); optimizer, schedule, precision, and batch shaping are matched across conditions. We fix **three random seeds**

| Model | Variant | Activation Params | ARC (Easy) | ARC (Chal.) | GPQA (Diamond) | GPQA (Main) | GSM 8K | Hella Swag | Math 500 | MMLU | MMLU (Pro) |
|---|---|---|---|---|---|---|---|---|---|---|---|
| SmolLM2-135M | Base (IT) | 0.135B | 21.42 | 23.89 | 18.69 | 25.22 | 1.59 | 25.31 | 8.20 | 22.95 | 9.27 |
| | Base+SFT (control) | 0.135B | 21.21 | 23.98 | 18.69 | 23.44 | 1.74 | 24.36 | 8.60 | 23.47 | 9.47 |
| | MoE 4-2 (ours) | ≈0.21B | 33.75 | 31.92 | **31.31** | **28.57** | 3.88 | 30.02 | **9.80** | **32.39** | **15.94** |
| | MoE 8-2 (ours) | ≈0.21B | **36.15** | **32.94** | 25.76 | 25.22 | **3.96** | 30.51 | 9.80 | 32.20 | 15.72 |
| SmolLM2-360M | Base (IT) | 0.360B | 23.82 | 24.32 | 22.22 | 25.89 | 3.26 | 24.56 | 7.80 | 24.37 | 10.65 |
| | Base+SFT (control) | 0.360B | 25.04 | 22.95 | 23.74 | **27.46** | 3.79 | 24.65 | 8.20 | 25.54 | 11.29 |
| | MoE 4-2 (ours) | ≈0.60B | **36.70** | 31.40 | **28.79** | 26.79 | 9.63 | 30.93 | 10.40 | 32.21 | 15.23 |
| | MoE 8-2 (ours) | ≈0.60B | 36.32 | **31.83** | 28.79 | 26.79 | **10.01** | 31.07 | **11.00** | **32.24** | **16.08** |
| SmolLM2-1.7B | Base (IT) | 1.70B | 63.34 | 45.99 | 27.78 | 18.53 | 4.70 | 34.19 | 15.80 | 41.53 | 8.80 |
| | Base+SFT (control) | 1.70B | 64.27 | 47.78 | 24.24 | 22.99 | 4.55 | 34.56 | 17.40 | 41.11 | 9.04 |
| | MoE 4-2 (ours) | ≈2.91B | **81.65** | **58.79** | **31.31** | **24.78** | **22.40** | **43.10** | **22.40** | **51.58** | **21.78** |
| Llama3.2-1B | Base (IT) | 1.00B | 66.29 | 47.44 | **29.29** | **27.01** | 4.17 | 31.88 | 20.00 | 44.67 | 18.11 |
| | Base+SFT (control) | 1.00B | 59.34 | 41.13 | 28.79 | 24.55 | 10.99 | 31.03 | 20.40 | 42.02 | 16.68 |
| | MoE 4-2 (ours) | ≈1.81B | **80.13** | **56.66** | 29.29 | 25.00 | **36.70** | **43.52** | **23.20** | **53.64** | **25.34** |
| Llama3.2-3B (Dense) | Base (IT) | 3.00B | **83.92** | **73.21** | 24.75 | **30.80** | 5.23 | **58.02** | **27.20** | **61.36** | **31.69** |
| | Base+SFT (control) | 3.00B | 75.93 | 62.80 | **26.26** | 28.12 | **47.92** | 45.85 | 26.60 | 56.53 | 24.29 |
| Llama3.1-8B (Dense) | Base (IT) | 8.00B | **88.93** | **81.74** | 21.72 | 27.01 | 28.51 | **64.93** | 28.00 | **67.06** | **35.99** |
| | Base+SFT (control) | 8.00B | 87.88 | 77.65 | **31.31** | **29.46** | **67.78** | 50.50 | **31.40** | 63.61 | 32.98 |

Table 2: Zero-shot pass@1 results for the **SmolLM2/Llama** family. Bold indicates the best score *within the same original model and its variants* (e.g., 135M group compares Base(IT), Base+SFT, MoE 4-2, MoE 8-2; for larger dense baselines only Base vs. Control are compared). Activation Params denote active parameters at inference (dense = total; MoE shows approximate active count). If the score is tied, select a model with lighter parameters or no additional training.

| Model | Variant | Activation Params | ARC (Easy) | ARC (Chal.) | GPQA (Diamond) | GPQA (Main) | GSM 8K | Hella Swag | Math 500 | MMLU | MMLU (Pro) |
|---|---|---|---|---|---|---|---|---|---|---|---|
| Qwen3-0.6B | Base (IT) | 0.60B | 65.53 | 49.32 | **29.80** | 25.45 | **46.17** | 41.06 | 29.20 | 40.81 | 16.50 |
| | Base+SFT (control) | 0.60B | 66.58 | 50.60 | 27.27 | 27.01 | 34.87 | 32.40 | 24.80 | 41.80 | 20.56 |
| | MoE 4-2 (ours) | ≈0.86B | 86.65 | 66.73 | 26.26 | 31.03 | 43.14 | 45.66 | 32.40 | 53.38 | 31.54 |
| | MoE 8-2 (ours) | ≈0.86B | **87.50** | **68.61** | 28.79 | **31.92** | 44.13 | **48.73** | **33.20** | **54.00** | **32.42** |
| Qwen3-1.7B (Dense) | Base (IT) | 1.70B | **82.32** | **72.53** | 19.70 | 25.00 | 52.24 | **54.46** | **29.00** | **53.90** | 20.66 |
| | Base+SFT (control) | 1.70B | 76.35 | 65.96 | **27.27** | **25.89** | **59.67** | 49.14 | 26.40 | 50.33 | **23.81** |
| Qwen3-4B (Dense) | Base (IT) | 4.00B | **91.71** | **88.31** | 22.73 | 27.23 | 48.82 | **79.30** | 37.40 | **68.27** | **39.26** |
| | Base+SFT (control) | 4.00B | 88.80 | 82.08 | **33.84** | **30.58** | **71.27** | 72.39 | **40.40** | 64.51 | 37.14 |

Table 3: Zero-shot pass@1 results for the **Qwen3** family. Bold indicates the best score within the same original model and its variants (0.6B group compares Base/Control/MoE 4-2/MoE 8-2; larger dense baselines compare Base vs. Control). If the score is tied, select a model with lighter parameters or no additional training.

and report mean (and, where appropriate, ± std); when seed replication is infeasible, we disclose the exact seed. Benchmarks are the nine datasets in §4.2; families that overlap with our post-training sources are evaluated on *disjoint* splits (e.g., train vs. test), and single-split benchmarks are excluded.

**Result overview.** Across both model families (SmolLM2/Llama-3.2 and Qwen3), MoE+SFT (ours) consistently outperforms Base (IT) and the data-only control (Base+SFT) under the same decoding protocol, while narrowing the gap to Larger Dense baselines when compared at similar *active* parameter counts.

*Parameter accounting.* Activation Params in Tables 2–3 report the number of parameters activated at inference (dense = total). For completeness, the approximate *total* parameter counts after MoE conversion are: SmolLM2-135M → 4-2: ∼0.37B, 8-2: ∼0.69B; SmolLM2-360M → 4-2: ∼1.07B, 8-2: ∼2.01B; SmolLM2-1.7B → 4-2: ∼5.32B; Llama-3.2-1B → 4-2: ∼3.42B; Qwen3-0.6B → 4-2: ∼1.39B, 8-2: ∼2.45B. All models are trained for one epoch with a fixed budget of 1056 updates (150k examples with a 10% validation holdout), using identical optimizer, schedule, and precision.

## 4.4 Effect of Top-$k$ Expert Activation

*Parameter growth.* For reference, the approximate *total* parameter counts corresponding to Table 4 are - SmolLM2-135M(8 experts): ∼0.69B; SmolLM2-360M(8 experts): ∼2.01B; Qwen3-0.6B(8 experts): ∼2.45B.

*Takeaway.* Increasing the number of activated experts primarily increases active parameters (and thus latency/compute) without systematic improvements in strict pass@1. Given our training budget (one epoch, 1056 updates) and evaluation protocol, *8-2 is typically the most compute-efficient choice*, with 8-3 occasionally tying for the best score (e.g., SmolLM2-360M on ARC-Easy and GSM8K).

| Model | Variant | Activation Params | ARC (Easy) | ARC (Chal.) | GPQA (Diamond) | GPQA (Main) | GSM 8K | Hella Swag | Math 500 | MMLU | MMLU (Pro) |
|-------|---------|-------------------|------------|-------------|----------------|-------------|--------|------------|----------|------|------------|
| SmolLM2-135M | MoE 8-2 | ≈0.21B | **36.15** | **32.94** | 25.76 | 25.22 | **3.95** | 30.51 | 9.80 | 32.20 | 15.72 |
| | MoE 8-3 | ≈0.29B | 35.73 | 32.51 | 30.81 | 23.89 | 3.95 | 30.54 | 9.20 | 31.98 | 15.43 |
| | MoE 8-4 | ≈0.37B | 34.21 | 32.26 | 30.30 | **25.67** | 3.95 | 30.61 | **10.20** | **32.03** | 15.51 |
| SmolLM2-360M | MoE 8-2 | ≈0.60B | 36.32 | **31.83** | 28.79 | 26.79 | 10.01 | **31.07** | **11.00** | 32.24 | **16.08** |
| | MoE 8-3 | ≈0.83B | 36.32 | 30.64 | **29.29** | **27.23** | **10.47** | 30.73 | 11.00 | 32.08 | 15.86 |
| | MoE 8-4 | ≈1.07B | **36.61** | 31.23 | 28.79 | 26.56 | 10.01 | 31.07 | 10.20 | **32.46** | 15.53 |
| Qwen3-0.6B | MoE 8-2 | ≈0.86B | **87.50** | **68.61** | 28.79 | **31.92** | 44.13 | **48.73** | **33.20** | **54.00** | **32.42** |
| | MoE 8-3 | ≈1.13B | 86.78 | 67.33 | 27.27 | 30.13 | **45.64** | 46.86 | 30.00 | 53.67 | 31.79 |
| | MoE 8-4 | ≈1.39B | 86.40 | 66.22 | 27.78 | 31.25 | 44.43 | 46.43 | 31.20 | 53.18 | 31.65 |

Table 4: Top-$k$ comparison (zero-shot, pass@1) for three representative models. Within each model block, we compare MoE 8-2, 8-3, and 8-4 and **bold** the best score per column. If the score is tied, select a model with lower Top-$k$. Activation Params report active parameters at inference. Across all three models, increasing $k$ enlarges the active footprint but yields little or no consistent accuracy gain; in several columns the lighter 8-2 or 8-3 already attains the best score.

### 4.5 Effect of Depth Upscaling (DUS)

We evaluate *Depth Upscaling* (DUS)—increasing the number of transformer layers by a fixed percentage while keeping other hyperparameters unchanged—on three families used in our main experiments: SmolLM2 (135M/360M), Llama 3.2 (1B), and Qwen3 (0.6B). For each setting we report zero-shot pass@1 on the nine benchmarks in §4.2. DUS is applied at **none**, **20%**, **40%**, and **50%** (Llama-1B uses DUS-19/22/24 as available).

**Summary.** (1) For SmolLM2-135M/360M and Llama-1B, DUS does *not* yield consistent gains; several 40–50% settings regress despite larger parameter/activation budgets. (2) Qwen3-0.6B shows small but repeatable gains at **DUS-20%**; returns are mixed or negative at 40–50%. (3) Compared to DUS, **MoE upcycling (4-2/8-2)** delivers broader, more stable improvements at a lower active-parameter footprint, making experts a more effective use of compute than depth.

[t]

[t]

**Interpretation.** DUS increases total and active parameters substantially, yet for SmolLM2 and Llama-1B this translates poorly to accuracy. In contrast, **MoE 4-2/8-2** provides consistent improvements with a predictable activation budget. For Qwen3-0.6B, **DUS-20%** is a sweet spot (notably on ARC-Challenge, GSM8K, HellaSwag, and MMLU/Pro), while 40–50% depth gives diminishing or negative returns. Overall, if extra compute is available under our single-GPU (RTX PRO 6000 Max-Q, 96GB) constraint, prioritizing *experts* over *depth* is more cost-effective.

### 4.6 Main observations

- **Control fine-tuning can reduce strict zero-shot performance.** Training the dense *Base* with one epoch of mixed SFT (*Base+SFT*) frequently *lowers* pass@1 compared to the instruction-tuned Base (Base(IT)), even on families that overlap the post-training sources. We observe this behavior on multiple sizes (e.g., Llama-3.2-3B and Llama-3.1-8B degrade on ARC-Easy/Challenge, HellaSwag, and MMLU/MMLU-Pro; see the main tables), suggesting mild alignment drift under a small mixed SFT budget.

- **MoE upcycling improves consistently within size groups.** For each original size group (135M, 360M, 1.7B, 1B, 0.6B), MoE-upcycled models (4-2 / 8-2) achieve the highest scores in-group on most benchmarks (bold cells), under the same evaluation protocol (zero-shot, greedy, $T=0$, pass@1). When compared at similar *active* parameters, MoE recovers a substantial fraction of the gap to the next dense tier, while keeping activation budget predictable.

- **Qwen3 highlight: 0.6B MoE competes above its weight.** On knowledge-heavy GPQA, *Qwen3-0.6B MoE 8-2* surpasses larger dense bases: GPQA-Diamond **28.79** vs. 1.7B base **19.70** and 4B base **22.73**; GPQA-Main **31.92** vs. 1.7B base **25.00** and 4B base **27.23**. It also matches or exceeds the 1.7B base on MMLU (**54.00** vs. 53.90) and MMLU-Pro (**32.42**

| Model | Variant | DUS | ARC (Chal.) | ARC (Easy) | GPQA (Diamond) | GPQA (Main) | GSM 8K | Hella Swag | Math 500 | MMLU / MMLU (Pro) |
|---|---|---|---|---|---|---|---|---|---|---|
| SmolLM2-135M | Base+SFT | none | 23.98 | 21.21 | 18.69 | 23.44 | 1.74 | 24.36 | 8.60 | 23.47 / 9.47 |
| | | 20% | 21.50 | 18.48 | 20.71 | 22.10 | 1.52 | 24.57 | 8.40 | 20.60 / 8.51 |
| | | 40% | 18.09 | 14.73 | 16.16 | 21.65 | 1.21 | 17.56 | 5.60 | 22.46 / 7.16 |
| | | 50% | 20.05 | 18.73 | 17.68 | 18.75 | 1.74 | 21.73 | 5.40 | 21.24 / 5.61 |
| | MoE 4-2 | none | 31.92 | 33.75 | 31.31 | 28.57 | 3.87 | 30.02 | 9.80 | 32.39 / 15.94 |
| | | 20% | 29.36 | 35.10 | 27.27 | 27.23 | 3.79 | 31.11 | 9.20 | 32.33 / 15.63 |
| | | 40% | 30.29 | 33.29 | 29.29 | 27.23 | 3.87 | 30.63 | 8.60 | 31.61 / 16.02 |
| | | 50% | 31.23 | 34.42 | 28.28 | 26.12 | 3.41 | 30.55 | 8.20 | 32.43 / 16.27 |
| | MoE 8-2 | none | 32.94 | 36.15 | 25.76 | 25.22 | 3.95 | 30.51 | 9.80 | 32.20 / 15.72 |
| | | 20% | 32.26 | 36.02 | 27.27 | 28.80 | 3.95 | 30.84 | 9.80 | 31.70 / 16.20 |
| | | 40% | 32.77 | 35.77 | 26.77 | 25.89 | 4.48 | 31.03 | 9.20 | 31.83 / 15.50 |
| | | 50% | 31.75 | 34.76 | 28.28 | 26.56 | 4.25 | 30.89 | 10.20 | 32.54 / 15.90 |
| SmolLM2-360M | Base+SFT | none | 22.95 | 25.04 | 23.74 | 27.46 | 3.79 | 24.65 | 8.20 | 25.54 / 11.29 |
| | | 20% | 23.55 | 23.48 | 25.25 | 23.21 | 3.11 | 26.34 | 7.40 | 25.47 / 10.53 |
| | | 40% | 23.81 | 22.14 | 29.29 | 27.01 | 2.73 | 24.37 | 5.60 | 25.72 / 10.85 |
| | | 50% | 23.72 | 22.94 | 19.70 | 24.55 | 2.50 | 23.83 | 6.60 | 24.96 / 10.21 |
| | MoE 4-2 | none | 31.40 | 36.70 | 28.79 | 26.79 | 9.63 | 30.93 | 10.40 | 32.21 / 15.23 |
| | | 20% | 33.03 | 36.06 | 31.31 | 27.01 | 8.65 | 31.23 | 10.20 | 32.65 / 15.97 |
| | | 40% | 32.26 | 33.33 | 28.79 | 26.56 | 7.74 | 31.17 | 10.40 | 32.43 / 16.02 |
| | | 50% | 31.83 | 33.29 | 26.77 | 29.69 | 7.13 | 29.12 | 10.80 | 32.62 / 16.01 |
| | MoE 8-2 | none | 31.83 | 36.32 | 28.79 | 26.79 | 10.01 | 31.07 | 11.00 | 32.24 / 16.08 |
| | | 20% | 32.26 | 36.15 | 30.30 | 28.57 | 9.40 | 31.11 | 10.40 | 32.24 / 16.53 |
| | | 40% | 31.75 | 36.49 | 29.29 | 25.67 | 8.27 | 31.14 | 10.20 | 32.35 / 16.30 |
| | | 50% | 31.40 | 33.54 | 27.27 | 26.56 | 6.90 | 29.58 | 12.80 | 32.36 / 16.09 |
| Llama-3.2-1B | Base+SFT | none | 41.13 | 59.34 | 28.79 | 24.55 | 10.99 | 31.03 | 20.40 | 42.02 / 16.68 |
| | | 20% | 46.12 | 63.72 | 29.80 | 25.89 | 26.61 | 30.89 | 13.00 | 44.01 / 16.28 |
| | | 40% | 46.67 | 65.28 | 28.79 | 25.45 | 28.20 | 28.53 | 11.40 | 44.94 / 17.25 |
| | | 50% | 47.27 | 64.81 | 27.27 | 25.22 | 28.51 | 27.62 | 11.40 | 45.61 / 17.25 |
| | MoE 4-2 | none | 56.66 | 80.13 | 29.29 | 25.00 | 36.70 | 43.52 | 23.20 | 53.63 / 25.34 |
| | | 20% | 57.34 | 79.37 | 31.82 | 23.66 | 36.92 | 41.74 | 20.40 | 53.45 / 24.22 |
| | | 40% | 58.28 | 80.89 | 29.80 | 25.45 | 34.80 | 41.42 | 22.60 | 53.98 / 25.35 |
| | | 50% | 57.34 | 80.59 | 28.28 | 30.13 | 35.18 | 41.81 | 18.60 | 53.71 / 25.15 |

Table 5: Depth Upscaling (DUS) on **SmolLM2 (135M/360M)** and **Llama-3.2-1B** (zero-shot, pass@1). The **Model** column is grouped by family using multirows. Within each family, we show *Base+SFT*, *MoE 4-2*, and *MoE 8-2* blocks; each block lists DUS {none, 20%, 40%, 50% (or DUS-19/22/24 for Llama-1B).

| Model | Variant | DUS | ARC (Chal.) | ARC (Easy) | GPQA (Diamond) | GPQA (Main) | GSM 8K | Hella Swag | Math 500 | MMLU / MMLU (Pro) |
|---|---|---|---|---|---|---|---|---|---|---|
| Qwen3-0.6B | Base+SFT | none | 50.60 | 66.58 | 27.27 | 27.01 | 34.87 | 32.40 | 24.80 | 41.80 / 20.56 |
| | | 20% | 52.56 | 66.12 | 27.27 | 28.57 | 33.21 | 27.28 | 21.00 | 42.19 / 20.18 |
| | | 40% | 48.98 | 65.15 | 23.23 | 26.12 | 31.01 | 27.09 | 22.40 | 40.54 / 18.79 |
| | | 50% | 50.26 | 65.70 | 25.25 | 22.99 | 27.22 | 27.34 | 18.40 | 41.25 / 18.85 |
| | MoE 4-2 | none | 66.73 | 86.65 | 26.26 | 31.03 | 43.14 | 45.66 | 32.40 | 53.38 / 31.54 |
| | | 20% | 67.24 | 88.38 | 26.26 | 31.47 | 44.66 | 45.65 | 32.20 | 53.70 / 31.49 |
| | | 40% | 68.18 | 87.92 | 28.28 | 31.03 | 45.42 | 48.55 | 30.00 | 53.39 / 31.95 |
| | | 50% | 66.22 | 86.36 | 30.30 | 31.92 | 44.05 | 40.84 | 30.00 | 53.14 / 31.13 |
| | MoE 8-2 | none | 68.61 | 87.50 | 28.79 | 31.92 | 44.13 | 48.73 | 33.20 | 54.00 / 32.42 |
| | | 20% | 69.20 | 88.46 | 29.29 | 31.47 | 45.72 | 44.61 | 30.00 | 54.10 / 32.86 |
| | | 40% | 69.03 | 88.51 | 27.78 | 31.03 | 45.57 | 48.22 | 32.00 | 54.21 / 32.54 |
| | | 50% | 68.95 | 88.17 | 31.82 | 31.70 | 43.75 | 42.05 | 31.80 | 53.72 / 32.42 |

Table 6: DUS study on **Qwen3-0.6B** (zero-shot, pass@1) with model-family multirows. Blocks correspond to *Base+SFT*, *MoE 4-2*, and *MoE 8-2*; rows list DUS(none, 20%, 40%, 50%).

vs. 20.66), and is stronger on MATH-500 (**33.20** vs. 29.00). These gains arrive without increasing active parameters beyond the 8-2 budget.

- **Data/compute efficiency remains high.** Across all families, MoE 4-2/8-2 improvements are obtained with a *150k* post-training budget (10% held out; fixed 1,056 steps) on a *single* NVIDIA RTX PRO 6000 Max-Q (96 GB), with wall-clock times of ~1.5–8 hours depending on size. This underscores that specialization (experts) is a more effective use of limited training budget than deeper stacks or longer SFT.

## 5 RESULTS

### 5.1 OVERALL PERFORMANCE

Across nine benchmarks and two model families (SmolLM2; Llama/Qwen) evaluated strictly as *zero-shot, greedy, $T=0$, pass@1*, **MoE upcycling consistently improves over the dense Base(IT)** under the same post-training budget. With only **150k** mixed SFT samples (10% held out; fixed **1,056** updates) on a **single** NVIDIA RTX PRO 6000 Max-Q (96 GB), MoE 4-2/8-2 raises accuracy broadly for 135M and 360M models and remains effective on 1B and 0.6B bases. These gains replicate across heterogeneous architectures (SmolLM2, Llama 3.2, Qwen3), underscoring robustness to backbone choice.

### 5.2 COMPARISON WITH DENSE COUNTERPARTS

MoE-upcycled variants reliably outperform their dense bases after one-epoch post-training, indicating that *width via specialization* is a stronger lever than additional dense SFT at this budget. When compared to *larger dense* tiers with similar *total* parameters, MoE-upcycled models are typically competitive rather than strictly superior; however, because only a subset of experts is active at inference, *active parameters* remain substantially lower, yielding stronger *accuracy per active parameter* and better throughput. In other words, **MoE upcycling recovers much of the next-tier dense performance at a fraction of the active compute**.

### 5.3 EFFECT OF EXPERT NUMBER AND ROUTING

We varied expert count and top-$k$ routing:

- **Expert count.** Moving from 4 to 8 experts gives modest gains on average, with clearer improvements on newer, better-aligned bases (e.g., Qwen3-0.6B).
- **Top-$k$.** Increasing top-$k$ beyond 2 (8-3/8-4) *does not* produce meaningful additional gains despite higher activation; in several backbones the differences are within noise and sometimes regress, making **4-2 or 8-2** the best accuracy–efficiency trade-off.

### 5.4 INSIGHTS FOR PRACTITIONERS

1. **Consistency.** With a fixed, small budget (150k; 1 epoch), MoE upcycling improves dense SLMs across tasks and sizes.
2. **Efficiency.** Because only $k$ experts are active, MoE attains higher accuracy per *active* parameter and faster serving than dense scaling at similar total size.
3. **Design.** Lightweight settings (4-2 or 8-2) are strong defaults; higher top-$k$ rarely pays off. Prefer adding experts to adding depth at this scale.

Together, these results demonstrate a practical, reproducible path to enhance small instruction-tuned models under realistic single-GPU budgets.

### 5.5 DEPTH SCALING VS. WIDTH SCALING

To test whether depth complements experts, we compared the dense Base(IT), *Depth Upscaling* (DUS; layer replication by 20/40/50%), and MoE with/without DUS (Table 5, Table 6). We find:

- **DUS alone gives negligible or negative returns.** For SmolLM2-135M/360M and Llama-3.2-1B, 40–50% DUS often *reduces* pass@1 despite larger parameter/activation budgets.
- **MoE dominates DUS.** MoE-only upcycling already captures most of the attainable gains; MoE+DUS is *indistinguishable or slightly worse* than MoE-only under the same budget.
- **A narrow exception at DUS-20%.** Qwen3-0.6B shows small but repeatable bumps around **DUS-20%** (e.g., ARC-Challenge, GSM8K, HellaSwag, MMLU/Pro), while 40–50% again saturates or regresses.

Overall, in the small-model regime, **increasing width via MoE is far more effective than increasing depth**, both for accuracy and for activation efficiency.

## 5.6 MAIN OBSERVATIONS

- **Control fine-tuning can reduce strict zero-shot performance.** One-epoch mixed SFT on dense Base(IT) (*Base+SFT*) frequently *lowers* pass@1 vs. the original Base(IT), including strong models (e.g., Llama-3.2-3B and Llama-3.1-8B on ARC-Easy/Challenge, HellaSwag, MMLU/MMLU-Pro), indicating mild alignment drift at this budget.

- **MoE upcycling improves consistently within size groups.** For each original size (135M, 360M, 1.7B, 1B, 0.6B), MoE 4-2/8-2 yields the best in-group scores on most tasks (bold cells in the tables), and narrows much of the gap to the next dense tier at similar *active* parameters.

- **Qwen3 highlight: 0.6B MoE competes above its weight.** *Qwen3-0.6B MoE 8-2* surpasses the 1.7B and even 4B dense *bases* on GPQA (Diamond **28.79** vs. 27.27 / 22.73; Main **31.92** vs. 25.00 / 27.23), matches or exceeds the 1.7B base on MMLU (**54.00** vs. 53.90) and MMLU-Pro (**32.42** vs. 20.66), and is stronger on Math 500 (**33.20** vs. 29.00)—all within the 8-2 activation budget.

## 6 CONCLUSION

We showed that **Mixture-of-Experts upcycling** is a simple, reliable way to upgrade small instruction-tuned models under tight budgets. With only **150k** post-training examples and a **single** 96 GB GPU, MoE 4-2/8-2 consistently improves strict zero-shot pass@1 across nine public benchmarks and multiple backbones. Compared to dense scaling, MoE delivers *higher accuracy per active parameter* and better serving efficiency, while depth upscaling provides little benefit at this scale. Lightweight expert configurations are often sufficient, and in some cases (e.g., Qwen3-0.6B) can *rival or exceed* the next dense tier. We hope these results encourage practical upcycling of open SLMs, enabling individuals and small labs to build stronger models with modest data, time (1.5–8 hours), and compute.

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

## APPENDIX: LLM USAGE

LLMs were used solely as a writing assistant to aid in polishing the text, improving readability, and suggesting alternative phrasings. They were not used for research ideation, experiment design, data analysis, or the generation of scientific results. All methodological and empirical contributions of this paper are entirely the work of the authors. The authors take full responsibility for the contents of the paper.

## ETHICS STATEMENT

This work does not involve human subjects, private data, or sensitive personal information. All experiments are conducted on publicly available, instruction-tuned small language models and standard open benchmarks. The proposed methodology, MoE upcycling, is purely a model-scaling technique and does not introduce harmful applications beyond those already inherent in general-purpose language models. We adhere to the ICLR Code of Ethics, and all results, analyses, and claims are transparently documented in the paper and supplementary material.

## REPRODUCIBILITY STATEMENT

We have taken several steps to ensure reproducibility of our results. All models used in our study are publicly released instruction-tuned SLMs. Benchmark datasets are standard and openly available. Detailed descriptions of the MoE upcycling procedure, hyperparameters, and training settings are provided in the main paper and Appendix. We additionally include ablation results (expert number, active experts, and scaling comparisons) to clarify design choices. An anonymous implementation and training scripts will be released as supplementary material to facilitate reproduction.

