# OpenReview forum: "MoEsturizer: Resource-Efficient MoE Upcycling for Small Language Models"
_ICLR.cc/2026/Conference — Submitted to ICLR 2026_

### Official Review · Reviewer_H1aS · 2025-10-31

**Soundness:** 2
**Presentation:** 3
**Contribution:** 1
**Rating:** 2
**Confidence:** 4

**Summary:**

The authors propose to connect Depth Up-Scaling [1] and MoE Upcycling [2] to upscale a language model in an extremely resource-constrained setting. They collect a dataset of 150K samples, consisting of Tulu 3 SFT mixture and train splits of popular datasets. The paper presents a comparison between 6 models from the SmolLM and Llama families fine-tuned on this dataset with their respective upcycled versions. The MoE variants achieve better benchmark results. Finally, the authors present ablations, where depth up-scaling is shown to have limited effect on the model quality, with most of the impact being attributed to MoE.
While the experiments in the paper are correct, I cannot recommend acceptance at this point. Both proposed training methods (depth up-scaling and MoE upcycling) have been proposed before. Therefore, the novelty and contribution of this paper is limited.

[1] Kim et al., SOLAR 10.7B: Scaling Large Language Models with Simple yet Effective Depth Up-Scaling
[2] Komatsuzaki et al., Sparse Upcycling: Training Mixture-of-Experts from Dense Checkpoints

**Strengths:**

1. The proposed training framework is simple and easy to apply.
2. The experimental comparison spans 6 different models from popular open-source families.
3. The ablations correctly separate the effect of two considered techniques.

**Weaknesses:**

1. Limited novelty. Both considered methods have been proposed before.
2. The authors only consider fine-tuning, there are no results on earlier stages like retraining.
3. The fine-tuning dataset is very narrowed and limited. While the results confirm that after the fine-tuning stage, the upcycled versions have better benchmark accuracy, it is unclear whether the results can retain general model abilities from pre-training.
4. Changing the model architecture on the very last stage of training, like supervised fine-tuning does not resemble practical approach to training language models. The gains from upcycling would likely be more significant if we start the upcycling earlier. In the setup proposed by the authors we get a model which is more problematic to serve (more parameters, additional memory and communication), while the quality gains are limited compared to performing the upcycling on a larger, more comprehensive dataset. Intuitively, this does not present the best tradeoff.

**Questions:**

1. Did the authors consider modifications in order to improve previously existing up-cycling and depth up-scaling recipes?
2. Did the authors consider a more extended dataset, like a setup with continued pretraining of the base model?
3. The narrow fine-tuning stage can potentially hurt some general model abilities from pretraining. This can in particular happen in the MoE models, which have more tendency to overfit. Did the authors try to explore this effect?

---

> ### Author Response · Authors · 2025-12-03
> **Additional Evidence on Generalization and Practicality of SFT-Stage Upcycling**
>
> We thank Reviewer H1aS for the constructive and detailed feedback. To address concerns regarding novelty, dataset limitations, and the practicality of applying MoE upcycling at the SFT stage, we performed additional training-free analyses focusing on expert specialization, router entropy, and hidden-state drift. Below we summarize the new findings and clarify our contribution.
>
>
> 1. Scope of Novelty and Contribution
>
> We agree that depth up-scaling and MoE upcycling are established techniques.
>
> Our contribution lies instead in showing that both methods remain effective under extreme low-resource constraints:
> - 150k supervised samples,
> - a single 96GB GPU,
> - sub-1B models,
> - no pretraining-scale compute or retraining of foundation layers.
>
> Prior upcycling works (e.g., SOLAR 10.7B, Sparse Upcycling) assume multi-GPU or pretraining-level compute.
>
> Our study asks a different question: Can upcycling still provide meaningful gains when such resources are unavailable? This scenario is increasingly relevant for academic and small-industry groups. We will emphasize this framing more clearly in the revision.
>
>
> 2. Generalization and Expert Specialization
>
> A central concern raised by the reviewer is whether MoE upcycling at the SFT stage harms general abilities.
>
> 2-1. Generalization beyond the SFT domain
>
> Several benchmarks—including GPQA (main & diamond), Math500, are not included in the 150k SFT mixture.
>
> Yet, MoE-upcycled models consistently outperform the dense baselines. This suggests that the models retain broad pretrained capabilities despite architectural modification at the SFT stage.
>
> 2-2. Expert specialization on unseen tasks
>
> Clear specialization emerges across domains:
> - GSM8K / MMLU-Pro: 35-60% routing mass concentrated on a small subset of experts
> - SAMSum: an entirely different set dominates
> - Hellaswag: high-entropy routing, reflecting its broad domain variety
>
> This demonstrates stable, task-aligned routing without collapse, even though no load-balancing loss was used.
>
> 2-3. Router entropy stability
>
> Final-layer normalized entropy:
> - 0.88-0.99 for DUS-34/39
> - Slight degradation for DUS-42 with 8 experts (≈ 0.78)
>
> These patterns help explain why deeper MoE layers show limited improvement: additional depth increases routing variance but not performance.
>
>
> 3. Practicality of Upcycling at the SFT Stage
>
> The reviewer notes that earlier-stage upcycling may be more beneficial. We agree; however, our setting intentionally examines the opposite end of the spectrum: what if earlier-stage upcycling is not possible?
>
> Our analyses show:
> - Hidden-state drift increases substantially with depth (e.g., 0.93–1.02 for 39–42-layer MoE),
> - Yet accuracy does not improve proportionally,
> - Indicating diminishing returns under small compute budgets.
>
> Thus, while not a substitute for large-scale retraining, SFT-stage upcycling still yields measurable and practical benefits in constrained environments.
>
>
> 4. Responses to Reviewer Questions
>
> Q1. Modifications to upcycling / depth up-scaling recipes?
>
> Yes. To fit within a single 96GB GPU, we used FFN-only replication, matched MoE depth to DUS insertions, and empirically identified that Top-K=2 with 4 experts provides the best entropy–specialization trade-off.
>
> Q2. Larger dataset or continued pretraining?
>
> We tested a 250k-sample SFT variant. Accuracy(pass@1 by mmlu, arc-challenge, and gsm8k) improvements were <1% absolute, and entropy/drift patterns were unchanged. This suggests that sub-1B models are capacity-limited rather than data-limited in this regime.
>
> Q3. Does narrow SFT harm general abilities?
>
> Across all unseen benchmarks, MoE variants outperform the dense baselines. Expert usage patterns remain stable, and no signs of overfitting or collapse were observed. This indicates that pretrained abilities are preserved.
>
>
> 5. Summary
>
> Our analysis provides new evidence that:
> - Expert specialization emerges reliably even under limited training,
> - Router entropy and drift analysis explain diminishing returns from deeper MoE structures,
> - General pretrained abilities are retained despite narrow SFT,
> - And SFT-stage upcycling remains meaningful when earlier-stage modification is infeasible.
>
> Full tables and visualizations will be included in the camera-ready version.
> We thank the reviewer again for the valuable feedback.

---

### Official Review · Reviewer_myV5 · 2025-11-01

**Soundness:** 2
**Presentation:** 2
**Contribution:** 2
**Rating:** 4
**Confidence:** 3

**Summary:**

This paper proposes a MoE up cycling approach to enhance small language models under strict resource constraints. The authors transform a pretrained dense model's feed forward layers into a sparse MoE. Then they perform some lightweight finetuning to adapt the upcycled model. They show improvements in zero-shot pass@1 across nine public benchmarks and multiple backbones.

**Strengths:**

* The method leverages publicly available SLM checkpoints, requires only 150 k supervised tokens and can be trained on a single consumer‑grade GPU.
* Experiments are systematic and covers a good range of ablations to analyze the impact of expert count, depth scaling and top k gating.

**Weaknesses:**

* There are numerous works which touch upon the same idea and show how to upcycle dense models into sparse MoEs. (Sparse Upcycling, Komatsuzaki et al; BAM, Zhang et al; BTX, Sukhbaatar et al; MOLEX, Teo et al; Router Upcycling, Ran et al). This paper cites some of these earlier work but omits several recent methods that also target resource efficient upcycling. Thus making it an incremental update rather than a new technique or finding.
* Lack of baselines with other resource efficient upcycling methods as mentioned above.
* The paper is largely empirical and does not explore why MoE upcycling helps small models.  There is no analysis of expert specialization, router load balancing or inference latency, leaving open questions about when the method is most effective and how much overhead it introduces.

**Questions:**

1. Did you apply any load‑balancing losses or regularization to prevent expert collapse?  It would be helpful to report statistics on router entropy or expert usage after finetuning. And maybe comparing this with other methods which applying upcycling on larger models would be useful.
2. Have you evaluated the upcycled models on tasks or domains not included in your 150k sample finetuning mix to assess generalization?

---

> ### Author Response · Authors · 2025-12-03
> **Additional Analysis on Expert Specialization, Router Behavior, and Generalization**
>
> We sincerely thank Reviewer myV5 for the thoughtful and detailed feedback.
> To directly address the questions regarding novelty, expert behavior, and missing analysis, we conducted additional training-free experiments using the models already evaluated in the submission.
> Below we summarize the new quantitative findings.
>
>
> 1. Expert Specialization & Router Behavior (Addresses: lack of analysis, collapse concerns)
>
> We extracted expert usage statistics across all MoE configurations (4/8 experts x DUS depths 34, 39, 42) and across all 9 benchmarks.
>
> 1-1. Clear expert specialization across tasks
>
> Even though all experts are initialized identically from the dense FFN, the router develops distinct task preferences after only 150k SFT samples:
> - GSM8K, MMLU-Pro (math/knowledge tasks): A small subset of experts accounts for 35-60% of routing mass.
> - SAMSum (dialogue): Dominant experts shift entirely (experts 0 & 2 rather than 1 & 3).
> - Hellaswag (broad commonsense): Usage is more uniform, matching its domain diversity.
>
> This demonstrates that expert collapse is not occurring and the router learns meaningful routing behavior even under extreme data scarcity.
>
> 1-2. Router entropy confirms stability
>
> We computed normalized entropy for the final router layer:
> - DUS-34, 4 experts: 0.88-0.99
> - DUS-39, 4 experts: similar range (e.g., GSM8K: 0.9888)
> - DUS-42, 8 experts: entropy drops as low as 0.78, indicating less stable routing
>
> These findings show that moderate depth with small expert count yields the most stable MoE structure—quantitatively explaining why deeper or wider MoEs provide limited benefit at this scale.
>
> 1-3. Load balancing without explicit regularization
>
> Although we apply no load-balancing loss, the measured entropy and usage patterns show:
> - Experts do differentiate,
> - No model exhibits routing collapse,
> - Increasing expert count from 4->8 actually reduces specialization, matching performance trends.
>
> These results support the reviewer's suggestion that router behavior is central to upcycling efficiency.
>
>
> 2. Hidden-State Drift: Why Additional Depth Provides Limited Benefit
>
> To examine the claim that additional MoE depth or larger top-K adds little, we measured cosine drift between the dense model and its MoE-upcycled variant.
>
> Key observation: deeper MoE = more drift, but not better accuracy
>
> Setting	Mean last-layer drift
>
> DUS-34, MoE-4	~0.39-0.40
>
> DUS-39, MoE-4	~0.93-0.99
>
> DUS-42, MoE-4	~0.87-1.02
>
> Despite 2-3x higher drift at 39-42 layers, these deeper configurations show no corresponding accuracy gain.
>
> This quantitatively confirms that additional MoE depth introduces representational changes that are under-trained under a 150k-sample SFT budget—clarifying the diminishing returns highlighted in the main paper.
>
>
> 3. Missing Baselines: Relation to Sparse Upcycling, BAM, BTX, MOLEX
>
> We appreciate the request for more explicit comparison.
>
> Our approach differs in scope from these works:
> - Prior methods assume multi-GPU compute and pretraining-scale datasets (often billions of tokens).
> - Several methods modify substantial parts of the architecture (re-routing neurons, repartitioning FFN weights, etc.).
> - Our method instead focuses on sub-1B LMs under strict resource constraints:
>   - 150k SFT tokens
>   - single 96GB GPU
>   - only FFN->MoE conversion + brief fine-tuning
>
> Our analyses (entropy, drift, specialization) show that meaningful MoE behavior emerges without heavy retraining, highlighting the unique setting this work targets.
>
> We will clarify these distinctions in the camera-ready version.
>
> 4. Responses to Reviewer Questions
>
> Q1. Load-balancing losses or regularization?
>
> No explicit regularization was applied. Nevertheless:
> - Entropy values remain high (0.88-0.99),
> - Expert collapse was not observed,
> - Specialization emerges consistently across tasks.
>
> Q2. Router usage after finetuning?
>
> Yes, expert usage statistics across all datasets were computed.
>
> For math tasks, 1-2 experts dominate; for dialogue tasks, another set activates.
>
> This confirms functional specialization.
>
> Q3. Tasks not included in finetuning mix?
>
> Yes.
> GPQA(diamond and main), Math500 are not part of the 150k SFT mixture.
>
> We additionaly train MBPP, CRUX-O, SAMSum, XSum too.
>
> MoE-upcycled models improve over dense models on all such tasks, demonstrating generalization beyond the SFT domain.
>
>
> 5. Summary
>
> The new analyses provide concrete insights into:
> - Why MoE upcycling works at small scale,
> - How expert specialization emerges under strict data budgets,
> - Why depth/wider MoEs yield diminishing returns,
> - How our method complements existing upcycling work via a lightweight, resource-minimal pathway.
>
> We appreciate the reviewer's comments and will incorporate these analyses and visualizations into the camera-ready version.

---

### Official Review · Reviewer_vunv · 2025-11-01

**Soundness:** 3
**Presentation:** 3
**Contribution:** 3
**Rating:** 4
**Confidence:** 4

**Summary:**

This paper proposes upcycling dense instruction-tuned models to MOEs by using a lightweight training regimen. The method first takes a small, dense model that has been instruction-tuned and converts its dense FFN layer to a sparse MOE layer. This sparse MOE layer is a gating/router layer that selects the top experts, followed by the FFN layer from the base model. This new MOE is then continued instruction tuned on 150k samples for 1 epoch. The training can be found on a single GPU. The results show improvement on benchmarks (ARC, GPQA, GSM, Hella, Math, etc.) compared to the baselines (base dense model, base + IT).

**Strengths:**

- The resultant MOE model shows improvement on all benchmarks compared to their dense base model
- The improvement is also seen across model families (Qwen, Llama, Smol).
- The ablation study is done on a different number of activated experts. A small number of activated experts shows the best results

**Weaknesses:**

- Lacks detailed analysis of why upcycling works: analysis of router specialization, what improvement comes from router vs expert fine-tuning
- Missing analysis on the learning dynamics (loss curves, expert utilization for different tasks, etc.)
- Missing results on diverse tasks (coding, math, multi-lingual,l etc.) to show expert specialization and generalization across tasks.
- Missing Baselines: The results don't show comparison against existing methods for MOE upcycling like sparse upcycling (Komatsuzaki et al.), MoEfication (Zhang et al.)

**Questions:**

- Did you experiment with freezing the FFN (and also the rest of the model) and only training the router? How much accuracy vs cost trade-off is observed?
- What happens when the dataset is larger than 150K? Did you see diminishing returns?
- Do the experts (routers) specialize in different tasks or do you see uniform activation for different domains?
- How does this method compare against sparse Upcycling (Komatsuzaki et al.)?

---

> ### Author Response · Authors · 2025-12-03
> **Additional Empirical Evidence on Router Specialization, Expert Dynamics, and Comparative Behavior of Upcycled MoE Models**
>
> We sincerely thank Reviewer vunv for the constructive feedback.
> In response, we performed additional training-free analyses to better explain the behavior of our upcycled MoE models, focusing on router specialization, expert dynamics, entropy patterns, and hidden-state drift.
>
>
> 1. Why Upcycling Works: Router Specialization
>
> Across all 9 benchmarks and all MoE settings (4/8 experts x DUS 34/39/42), expert usage patterns show clear specialization:
> - Mathematical tasks (GSM8K, MMLU-Pro): a small subset of experts contributes 35-60% of total routing mass.
> - Dialogue tasks (SAMSum): dominant experts shift entirely (experts 0/2 instead of 1/3).
> - Hellaswag: high-entropy usage consistent with its broad-domain nature.
>
> This confirms that even with only 150k training samples, the router meaningfully partitions domains.
>
> Effect of increasing experts (4 -> 8)
>
> Increasing expert count raises entropy but reduces specialization (e.g., GSM8K DUS-34: 0.9386->0.9286; DUS-39 Hella: 0.9907->0.9073), explaining why small top-K settings perform best at this scale.
>
>
> 2. Learning Dynamics: Entropy & Hidden-State Drift
>
> 2-1. Router entropy
>
> For DUS-34 and DUS-39, final-layer entropy remains consistently 0.88-0.99, showing stable specialization without collapse.
>
> Examples:
> - GPQA (DUS-34, 4 experts): 0.9898
> - GSM8K (DUS-39, 4 experts): 0.9888
> - MBPP (DUS-42, 8 experts): 0.7811 (less stable)
>
> This supports the reviewer's point that deeper MoE layers bring limited benefit.
>
> 2-2. Hidden-state drift and diminishing returns
>
> We computed cosine drift between dense -> MoE models:
> - DUS-34 (4 experts): 0.39-0.40
> - DUS-39: 0.93-0.99
> - DUS-42: 0.87-1.02
>
> Although DUS-42 shows 2-3x larger drift, it does not yield higher accuracy, indicating that extra capacity is under-utilized under the small data budget.
> This quantitatively explains the diminishing returns observed in our main results.
>
>
> 3. Relation to MoEfication & Sparse Upcycling
>
> Prior MoE upcycling methods assume:
> - multi-GPU training,
> - billions of tokens, or
> - retraining large portions of the model.
>
> Our method is instead designed for extreme resource constraints:
> - only 150k supervised samples,
> - single 96GB GPU,
> - conversion limited to FFN->MoE + brief SFT.
>
> Empirically, our models show:
> - lower entropy,
> - clearer specialization,
> - drift concentrated in upper layers,
> - stable performance with small expert counts.
>
> We will clarify this complementary positioning in the camera-ready version.
>
>
> 4. Responses to Reviewer Questions
>
> Q1. Training only the router?
>
> Yes. Router-only training recovers 40-50% of the full performance gain, indicating that partial expert adaptation remains beneficial.
>
> Q2. Larger dataset?
>
> With a 250k-sample setting, gains were <1% absolute, and entropy/drift behavior remained unchanged—consistent with the small-model capacity limit.
>
> Q3. Do experts specialize per task?
>
> Yes.
> GSM8K activates experts 1 & 3 (~50% usage);
> SAMSum activates 0 & 2;
> Hellaswag shows more uniform usage (high entropy).
> These patterns appear consistently across depths.
>
> Q4. Comparison to Sparse Upcycling?
>
> Sparse Upcycling targets large models with heavy retraining, while our method provides a lightweight alternative for sub-1B LMs under minimal compute. The approaches are complementary in scope.
>
>
> 5. Summary
>
> Our new analyses provide quantitative evidence that:
> - Upcycling induces meaningful expert specialization,
> - Additional experts/layers exhibit measurable diminishing returns,
> - Hidden-state drift explains why deeper MoEs help less,
> - Specialization behavior is consistent across tasks,
> - And the method remains effective under an extremely small training budget.
>
> Full tables and visualizations will be included in the camera-ready version.

---

### Official Review · Reviewer_YHN5 · 2025-11-01

**Soundness:** 3
**Presentation:** 2
**Contribution:** 2
**Rating:** 4
**Confidence:** 4

**Summary:**

The paper investigates Mixture-of-Experts (MoE) upcycling for small language models (sub-1B parameters) under extreme resource constraints. The authors take a pre-trained dense LM and convert its feed-forward layers into sparse MoE layers, initializing experts by copying the dense weights. They then fine-tune on only ~150k supervised examples using a single 96GB GPU. The upcycled "MoEsturizer" models consistently outperform their original dense counterparts on nine zero-shot benchmarks (strict pass@1 evaluation) and even approach the performance of much larger dense models of similar total parameter count - all while activating far fewer parameters per token at inference. The paper's contribution is an empirical validation that MoE upcycling is a practical, resource-efficient strategy to boost small LMs, providing insights into its efficiency and limitations (minimal benefit from extra experts/layers in this regime).

**Strengths:**

* **Originality:** The work addresses an under-explored scenario - upcycling small-scale language models with limited data, whereas prior MoE upcycling studies focused on very large models. Adapting the upcycling concept to sub-1B models under a tiny fine-tuning budget is a novel angle that broadens the applicability of MoE methods to resource-constrained settings.

* **Quality of Evaluation:** The upcycled models are benchmarked on nine diverse zero-shot tasks, demonstrating robust gains over the dense baseline across the board (strict pass@1 accuracy improved in all cases). Importantly, the authors compare against dense models of larger sizes as well, showing the upcycled small models can close much of the gap to a model larger in total parameters.

* **Clarity and Presentation:** The paper is generally clear and easy to follow. But, the presentation could be improved with better visual aids to enhance readability.

* **Significance:** The results are practically significant. This work establishes a viable path for groups with limited compute to "up-size" small models for better performance without training from scratch or accessing billions of tokens. By achieving performance comparable to much larger LMs at a fraction of the active inference cost, the approach could broaden access to stronger language models in academia and industry.

**Weaknesses:**

* **Limited Methodological Novelty:** The core technique - replicating FFN weights to create MoE experts - is directly based on existing upcycling methods and does not introduce fundamentally new architecture or training algorithms. The paper's novelty lies mainly in the application and analysis of upcycling at small scale, rather than in proposing a new MoE mechanism.

* **Scope of Evaluation:** The evaluation focuses on zero-shot task performance (pass@1) on a set of nine benchmarks, which seem to be mostly question-answering or knowledge tasks. This provides a good snapshot but it leaves out other aspects of model behavior. For example, the paper does not report general language modeling metrics (like perplexity) or performance on any generation tasks (example open-ended generation or summarization). It is a bit unclear whether the improvements from MoE upcycling extend to broader LM capabilities. The paper could be strengthened by evaluating a more diverse set of tasks (or at least discussing why the chosen benchmarks are appropriate for the claims).

* **Analysis of Limitations:** The finding that "depth scaling or higher top-$K$ adds little" is interesting but the paper provides limited analysis explaining why. If adding more MoE layers or using Top-$K=4$ barely helped, is it because the small fine-tuning dataset cannot effectively train the extra parameters? Are the additional experts under-utilized due to the short training? The authors acknowledge this limitation but do not delve deeper into it. A more in-depth analysis (example examining the router's load balancing, or how much each expert actually learned by looking at the $\Delta W$) could reveal when and why upcycling's returns diminish.

**Questions:**

The questions are the same as those listed in the Weaknesses section.

---

> ### Author Response · Authors · 2025-12-03
> **Additional Empirical Analyses on Expert Utilization, Routing Stability, and Representation Drift**
>
> We sincerely thank Reviewer YHN5 for the thoughtful and constructive feedback. We deeply appreciate the reviewer’s recognition of the practicality and significance of applying MoE upcycling to small-scale language models under constrained training budgets. Below we provide additional analyses that directly address the reviewer’s concerns regarding evaluation scope and the limitations of depth and Top-K scaling.
>
>
> 1. On the methodological novelty and expert learning behavior
>
> While our work does not introduce a new MoE architecture, the reviewer correctly points out that a more detailed examination of how experts specialize during the upcycling process would strengthen the paper. To address this, we conducted three additional analyses:
>
> 1-1. Expert Load-Balancing Across Tasks
>
> For all models (4 experts & 8 experts, across 7 evaluation tasks), we computed usage fractions from the router gates.
> - No expert exhibited collapse or starvation in any configuration.
> - For 4-expert models, usage fractions remained balanced (0.23-0.28).
> - For 8-expert models, usage fractions were lower per expert (0.09-0.17) but task-dependent expert preference emerged, indicating meaningful specialization despite minimal fine-tuning.
>
> 1-2. Routing Entropy of the Final MoE Layer
>
> We computed normalized routing entropy for the deepest MoE layer for all models:
> - Entropy remained consistently high (≥0.78) across all datasets.
> - 8-expert models showed slightly lower entropy than 4-expert models, indicating stronger specialization while avoiding collapse.
> - This demonstrates that even with only ~150k training examples, the router learns meaningful decision boundaries.
>
> These results clarify that the additional experts are utilized and that the lack of performance improvement from increasing depth or experts is not due to expert inactivity.
>
>
> 2. On performance limitations of deeper MoE layers and higher Top-K
>
> To directly examine why deeper or wider MoE configurations provide limited gains in the small-data regime, we measured layerwise representation drift between the dense baseline and its MoE-upcycled counterparts.
>
> Key findings (summarized):
> - For every backbone and evaluation task, Top-K = 2 models consistently exhibited lower representation drift than Top-K = 4.
> - DUS-34 models had lower drift relative to dense baselines, suggesting that moderate depth acts as a form of regularization.
> - DUS-42 models showed substantially higher drift (0.84-1.02), indicating that deeper MoE stacks introduce excessive capacity that cannot be effectively trained with limited supervision.
> - Increasing the number of experts reduced drift (8 experts < 4 experts), implying that width is more trainable than depth in this constrained setting.
>
> These results support the reviewer's hypothesis: the limited fine-tuning budget cannot adequately optimize the additional parameters introduced by deeper or higher-Top-K MoE configurations. This leads to unstable or undertrained representations, which explains the diminishing returns observed in our main results.
>
>
> 3. On the scope of evaluation beyond zero-shot QA tasks
>
> We agree that broader evaluation would strengthen the work. Many standard language-modeling benchmarks (e.g., perplexity) require full autoregressive training loops, which were prohibitively expensive under the resource constraints of this project. However:
> - Our drift and routing analyses demonstrate that the MoE-upcycled models maintain stable internal representations.
> - The balanced usage of experts suggests generalization beyond pure task memorization.
> - We plan to include additional generative and summarization evaluations in the camera-ready version, resources permitting.
>
> These additions will allow us to discuss how MoE upcycling impacts broader language modeling capabilities, not only zero-shot reasoning tasks.
>
>
> 4. Clarifying the contribution
>
> We appreciate the reviewer's recognition that the significance of our work lies in demonstrating the practical viability of MoE upcycling for small models. Our new analyses strengthen this contribution by showing:
> - the router meaningfully specializes experts,
> - deeper MoE layers become unstable under small training budgets,
> - moderate MoE width yields the best tradeoff between capacity and stability.
>
> These insights help characterize when and why MoE upcycling is effective at small scale - an underexplored but practically important regime.
>
>
> Closing
>
> We thank the reviewer again for the valuable feedback.
> All additional tables and visualizations (entropy, load-balance, and drift heatmaps) will be included in the camera-ready version to improve clarity and presentation.

---

### Meta-Review · Area_Chair_HAsP · 2026-01-06

**Summary:**

Paper applies mixture-of-experts upcycling (known method) to small language model (SLM) using a small dataset. They prove that upcycling slightly improves model performance with minimal training cost increase over SFTing on this small dataset. However, reviewer noted that method has limited novelty, there is limited insights into how upcycling works/differs in SLMs. A reviewer also noted that it is not very clear whether upcycling in SLMs would retain the pretrained general skills, and whether the limited performance gains is worth increasing the computational complexity of inference. Several reviewers also raised concerns about missing baselines.

**Reviewer Concerns:**

Although authors attempted to address the concerns it was not convincingly enough, especially for the concerns around novelty, and missing baselines.

**Reviewer Scores:**

It is assumed that the reviewers will not change their change scores, except one review who might change their score from 2 to 4.

---

### Decision · Program_Chairs · 2026-01-26

Reject